# The effect of game-based education on adherence to treatment and anxiety level in type 2 diabetics started on insulin therapy

Esin Erdem[1], Gönül Düzgün[2]*

1 Internal Medicine Nursing, Uşak Training and Research Hospital, Diabetes Education Unit, Uşak, Türkiye, 2 İzmir Tınaztepe University Vocational School of Health Services First and Emergency Aid Program İzmir, Türkiye

* gonul.duzgun@tinaztepe.edu.tr

## Abstract

### Background

Insulin initiation in type 2 diabetes mellitus (T2DM) is often hindered by anxiety and poor treatment compliance. Although game-based learning may enhance patient engagement, evidence regarding its effectiveness in adult populations remains limited.

### Objective

To evaluate the effect of a game-based education program on treatment compliance (primary outcome) and anxiety levels (secondary outcome) in adults with T2DM receiving insulin therapy.

### Methods

This randomized controlled study included 72 adults with T2DM who were assigned to either a game-based education group (n = 36) or a standard lecture-based education group (n = 36). Treatment compliance was assessed using the Patient Compliance Scale for T2DM Treatment (primary outcome), and anxiety was measured using the Beck Anxiety Inventory (secondary outcome). Between-group differences were analyzed using baseline-adjusted ANCOVA, controlling for baseline scores, age, duration of diabetes, and employment status. Exploratory correlation analyses examined the association between anxiety and treatment compliance.

### Results

Baseline demographic and clinical characteristics were comparable between the intervention and control groups (p > 0.05). Following the intervention, participants in the game-based education group demonstrated significantly higher treatment compliance compared with those in the control group. Baseline-adjusted ANCOVA

**Data availability statement:** Data cannot be publicly shared due to ethical restrictions involving sensitive patient information. De-identified data are available upon reasonable request from the Uşak University Non-Interventional Clinical Research Ethics Committee (contact: kaek@usak.edu.tr), subject to institutional approval.

**Funding:** The author(s) received no specific funding for this work.

**Competing interests:** none.

revealed a significant between-group difference favoring the intervention group for total treatment compliance (F = 65.92, p < 0.001), indicating a substantial improvement associated with the intervention.

Adjusted analyses also showed significantly lower post-intervention anxiety scores in the intervention group compared with the control group (F = 11.241, p < 0.001), with a large effect size (partial $\eta^2$ = 0.14). Improvements were observed across most treatment compliance sub-dimensions, with the exception of lifestyle change, which did not reach statistical significance.

Exploratory correlation analyses further indicated that the negative association between anxiety and treatment compliance weakened following the intervention only in the game-based education group, whereas this relationship remained significant in the control group.

## Conclusion

Game-based education is more effective than standard lecture-based education in enhancing treatment compliance and reducing anxiety among individuals initiating insulin therapy. Additionally, the intervention may attenuate the adverse impact of anxiety on adherence-related behaviors.

## Trial registration

ClinicalTrials.gov NCT07195188

## Introduction

Diabetes mellitus is one of the most important global public health problems and is defined by the World Health Organization (WHO) as a non-communicable epidemic disease [1]. The global prevalence of diabetes continues to rise, leading to serious complications that negatively affect quality of life and impose a substantial burden on healthcare systems. According to the International Diabetes Federation (IDF), the number of individuals living with diabetes is expected to increase significantly in the coming decades [2].

Type 2 diabetes mellitus (T2DM) constitutes the majority of diabetes cases and is characterized by insulin resistance and progressive pancreatic β-cell dysfunction. Effective diabetes management requires sustained engagement in healthy nutrition, regular physical activity, appropriate medical treatment, and long-term self-care behaviors [3]. It has been reported that nearly all diabetes management outcomes depend on patients' self-care practices, underscoring the importance of acquiring adequate knowledge, skills, and motivation to manage this chronic condition [4]. However, treatment adherence can be particularly challenging during complex therapeutic transitions, such as the initiation of insulin therapy. Fear of injections, insufficient knowledge, and psychological distress frequently accompany this period and may lead to reduced adherence and increased anxiety [5].

Treatment adherence plays a central role in achieving optimal glycemic control, preventing complications, and improving quality of life in individuals with diabetes. The WHO emphasizes that improving adherence to treatment may have a greater impact on population health outcomes than advances in specific medical treatments alone. In patients with T2DM who have recently initiated insulin therapy, adaptation to treatment often represents a critical and vulnerable phase. Difficulties in adhering to insulin regimens may not only compromise individual health outcomes but also increase healthcare utilization and costs [6].

In the literature, terms such as *treatment adherence* and *patient compliance* are sometimes used interchangeably; however, they represent distinct concepts. Treatment adherence generally refers to the extent to which patients' behaviors—such as medication-taking and lifestyle practices—align with agreed therapeutic recommendations and emphasizes patients' active role in disease management. Patient compliance, in contrast, reflects a more passive following of medical instructions. More recently, broader constructs such as *patient engagement* or *patient activation* have been proposed, encompassing behavioral, cognitive, and emotional involvement in care. The present study focuses primarily on the behavioral dimension of diabetes management, operationalized as treatment adherence, while acknowledging that cognitive and emotional components of patient engagement remain important areas for future research.

Diabetes education is widely recognized as a cornerstone of effective disease management. Numerous studies have demonstrated that structured education programs can enhance diabetes-related knowledge, improve treatment adherence, and positively influence glycemic outcomes [6–8]. Nevertheless, some educational interventions fail to produce sustained behavioral change, suggesting that both educational content and teaching methods are critical determinants of effectiveness [5,9]. In this context, game-based educational approaches—including gamification strategies, digital games, video animations, game consoles, and board games—have gained increasing attention in diabetes education [10–16]. Despite this growing interest, empirical evidence regarding the effectiveness of game-based education, particularly among adults with T2DM initiating insulin therapy, remains limited.

Game-based learning environments promote active participation, immediate feedback, and experiential learning, which may enhance motivation and retention of information. Game elements such as interaction, challenge, and repetition can facilitate understanding of complex health-related concepts and support behavioral change [17,18]. In healthcare settings, game-based education has been shown to increase patient motivation, reduce stress and anxiety, and support adherence to treatment recommendations. These benefits may be especially relevant during emotionally sensitive transitions, such as the initiation of insulin therapy, where fear and anxiety can interfere with effective self-management [19].

The primary objective of this study was to evaluate the effect of a game-based educational intervention on treatment adherence in adults with type 2 diabetes who had recently initiated insulin therapy. Secondary objectives included examining the effect of the intervention on anxiety levels. To this end, the board game *"Let's Learn Diabetes"* was administered to individuals who had been receiving insulin therapy for a maximum of three months. Treatment adherence and anxiety levels were assessed before and after the intervention in the game-based education group and compared with those of a control group receiving standard presentation-based education.

## Study design

This experimental study was conducted using a randomized, pre-test–post-test, parallel-group controlled design. The primary objective was to evaluate the effect of a game-based educational intervention on **treatment adherence** in adults with type 2 diabetes who had recently initiated insulin therapy. Secondary objectives included examining the effect of the intervention on anxiety levels. The study was registered at ClinicalTrials.gov (Identifier: NCT07195188).

## Study population and sample

The study population consisted of individuals with type 2 diabetes who had initiated insulin therapy within the previous three months and who attended the Internal Medicine and Endocrinology outpatient clinics and diabetes education unit

of a training and research hospital between May and October 2024. Eligible individuals who met the inclusion criteria and agreed to participate were consecutively enrolled.

A total of 79 patients were assessed for eligibility. Seven declined participation, and 72 participants were included in the final analysis. Participants were randomly assigned to either the intervention group (n = 36) or the control group (n = 36) (Fig 1).

## Sample size calculation

The sample size was calculated based on the primary outcome, defined as the total score of the Patient Compliance Scale for Type 2 Diabetes Mellitus Treatment. As no prior randomized controlled trials using this specific scale in a game-based education context were available, the expected effect size was estimated using a conservative approach informed by previous behavioral and educational interventions in diabetes management.

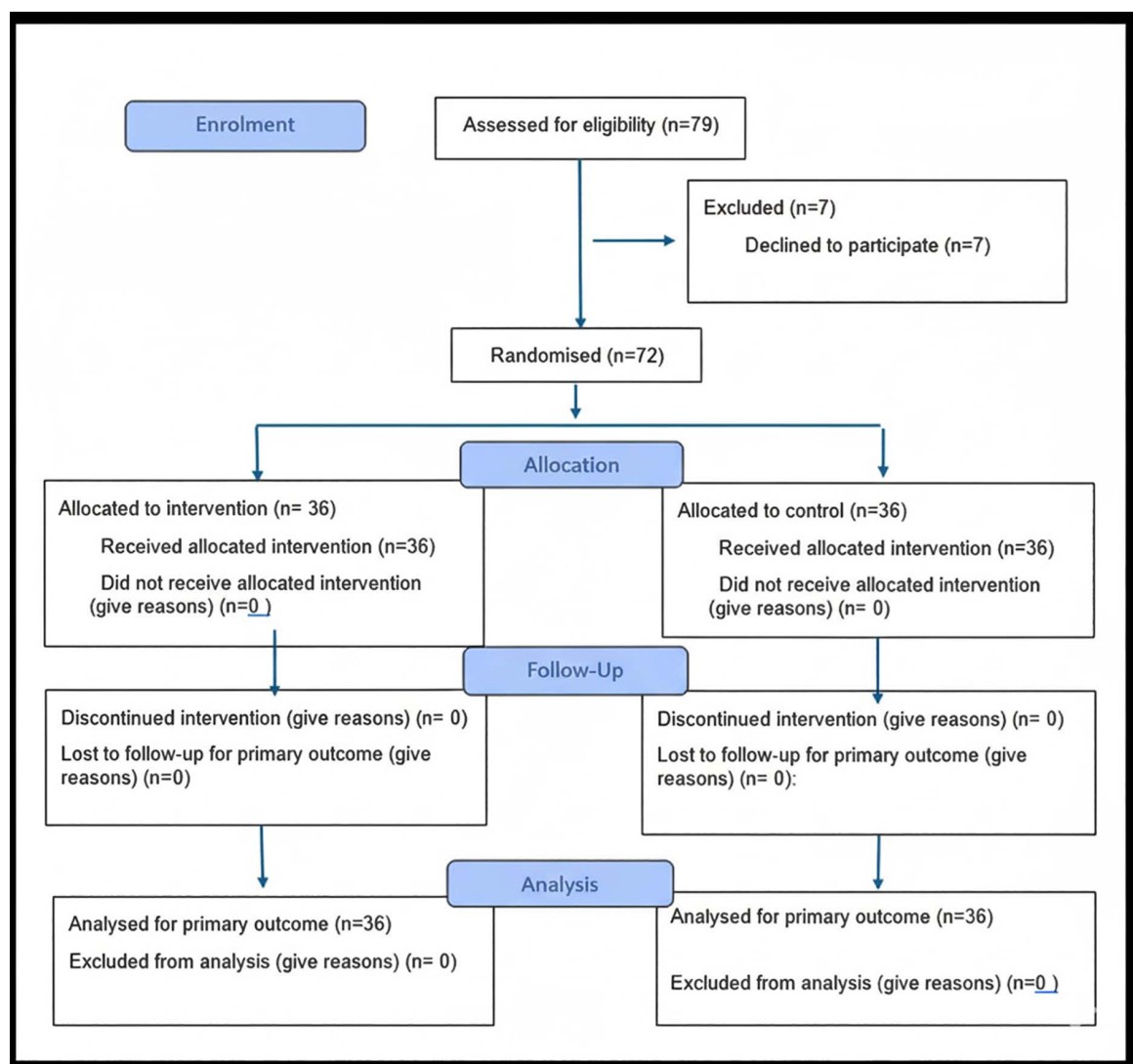

**Fig 1. CONSORT participant flow diagram.**

A medium-to-large standardized effect size (Cohen's d = 0.70) was assumed for the between-group difference in post-intervention treatment compliance. Using a two-sided significance level of 0.05 and a statistical power of 80%, a minimum of 32 participants per group was required based on a two-sample t-test approximation.

Although the primary analysis was planned using baseline-adjusted analysis of covariance (ANCOVA), the t-test–based sample size estimation was considered appropriate and conservative, as ANCOVA generally provides greater statistical power when baseline measurements are included. To account for potential attrition, the target sample size was increased to 36 participants per group.

## Randomization and allocation

Participants were randomly allocated to the intervention or control group using a computer-generated random sequence with a 1:1 allocation ratio. Randomization was conducted by an independent researcher who was not involved in participant recruitment, data collection, or intervention delivery. Group assignments were concealed until baseline assessments were completed.

## Inclusion and exclusion criteria

**Inclusion criteria** were: diagnosis of type 2 diabetes for at least one year; initiation of insulin therapy within the previous three months; age between 40 and 65 years; absence of diagnosed psychiatric disorders; and willingness to participate.

**Exclusion criteria** included: diagnosis of diabetes for less than one year; insulin therapy for longer than three months; presence of mental or psychological comorbidities; age outside the specified range; refusal to participate; or non-attendance at one or more of the four scheduled educational sessions.

## Data collection ınstruments

Data were collected using a Patient Identification Form, the Beck Anxiety Scale (BAS), and the Patient Compliance Scale for Type 2 Diabetes Mellitus Treatment.

The BAS is a 21-item self-report instrument assessing anxiety symptoms on a 4-point Likert scale, with higher scores indicating greater anxiety severity. The Turkish validity and reliability were established by Ulusoy et al. In the present study, Cronbach's alpha coefficients were 0.946 (pre-test) and 0.955 (post-test).

Treatment adherence was assessed using the Patient Compliance Scale for Type 2 Diabetes Mellitus Treatment, a 30-item Likert-type scale developed for the Turkish population. Total scores range from 30 to 150, with lower scores indicating higher adherence. The scale includes seven sub-dimensions related to attitudes, knowledge, lifestyle behaviors, emotional responses, and denial. Cronbach's alpha values in this study were 0.791 at baseline and 0.918 post-intervention.

## Intervention procedures

Following baseline assessments, participants attended four weekly educational sessions lasting approximately two hours each.

**Intervention Group:** Participants received education through the *"Let's Learn Diabetes"* board game, a game-based educational tool developed by the researchers (Turkish Patent Office Design Registration No: 2023 013290). The game incorporates competition, teamwork, feedback, and progressive difficulty levels to support active learning. Each session involved structured gameplay facilitated by a trained nurse, focusing on diabetes knowledge, insulin administration, self-monitoring, nutrition, exercise, and complication prevention.

**Control Group:** Participants received standardized lecture-based diabetes education covering the same content areas as the intervention group, delivered through traditional presentations and question–answer sessions.

Post-test assessments were conducted approximately one month after completion of the educational sessions.

## Outcomes

The primary outcome was treatment adherence, measured by the total score of the Patient Compliance Scale post-intervention, adjusted for baseline values. Secondary outcomes included anxiety levels measured by the BAS and exploratory analyses of adherence subscale scores.

## Statistical analysis

Statistical analyses were performed using SPSS version 25.0. Descriptive statistics were calculated for baseline demographic and clinical characteristics. Between-group differences in the primary outcome, treatment compliance (total Patient Compliance Scale score), and secondary outcomes, including anxiety levels and compliance sub-dimensions, were assessed using analysis of covariance (ANCOVA). All ANCOVA models were adjusted for the corresponding baseline score, age, duration of diabetes, and employment status. Adjusted mean differences with 95% confidence intervals, F statistics, and p values were reported.

Exploratory analyses involving compliance sub-dimensions and correlation analyses were interpreted cautiously. Statistical significance was set at $p < 0.05$.

## Ethics statement

The study was conducted in accordance with the Declaration of Helsinki and approved by the Uşak University Non-Interventional Clinical Research Ethics Committee (Decision No: 352-352-10). Written informed consent was obtained from all participants prior to data collection.

## Findings

### Reliability and distribution of the scales

Cronbach's alpha coefficients greater than 0.60 indicated acceptable internal consistency for all scales used in the study (Table 1). The total scores of both the Patient Compliance Scale for Type 2 Diabetes Treatment and the Beck Anxiety Scale demonstrated high internal consistency at both time points. While several subscales showed low Cronbach's alpha values at baseline, reliability coefficients improved notably after the intervention. Subscales with alpha values below the

Table 1. Internal consistency (Cronbach's alpha) of study instruments and subscales at baseline and post-intervention.

| Scale/ Subscale | Pre-test | Post-test |
|---|---|---|
| Patient Compliance Scale in Type 2 Diabetes Treatment (Total) | 0.791 | 0.918 |
| Attitudes and emotional factors | 0.757 | 0.821 |
| Knowledge and personal factors | 0.707 | 0.747 |
| Lifestyle change | 0.612 | 0.735 |
| Feelings of anger | 0.392 | 0.591 |
| Emotions and behaviors appropriate to sleep | 0.160 | 0.626 |
| Diet bargaining | 0.366 | 0.607 |
| Sense of denial | 0.127 | 0.659 |
| Beck Anxiety Scale | 0.946 | 0.955 |

Cronbach's alpha values ≥0.60 were considered acceptable. Subscales with lower alpha values at baseline were retained for exploratory analyses and interpreted with caution.

acceptable threshold were retained for exploratory purposes, and interpretations related to these subscales were made cautiously.

The intervention and control groups were largely comparable with respect to sex, age, marital status, education level, diabetes duration, treatment type, insulin use characteristics, and diabetes-related complications ($p > 0.05$ for all) (Table 2).

A statistically significant difference was observed only in employment status between the groups ($p = 0.043$). Given its potential prognostic relevance, employment status was considered as a covariate in the adjusted between-group analyses. Overall, the similarity of baseline characteristics supports the adequacy of the randomization procedure.

A statistically significant between-group difference in post-test anxiety scores was observed, with higher anxiety levels in the control group compared to the intervention group. Within-group analyses indicated significant reductions in anxiety scores from pre-test to post-test in both groups.

Baseline-adjusted ANCOVA demonstrated a significant between-group difference in post-test anxiety scores after controlling for baseline anxiety, age, and duration of diabetes ($F = 11.241$, $p < 0.001$, partial $\eta^2 = 0.14$). The intervention group exhibited significantly lower adjusted post-test Beck Anxiety Inventory scores than the control group, indicating a clinically meaningful reduction in anxiety following game-based diabetes education (Table 3).

At baseline, there were no statistically significant differences between the groups in total treatment compliance scores or in most subscale scores ($p > 0.05$), indicating comparable baseline levels prior to the intervention.

Following the intervention, statistically significant between-group differences were observed in the post-test total Patient Compliance Scale scores, with the control group showing higher scores compared to the intervention group ($p < 0.001$), indicating poorer treatment adherence given that higher scores reflect lower compliance. Although raw post-test scores were numerically higher in the control group, baseline-adjusted ANCOVA analyses favored the intervention group, demonstrating greater improvements in treatment compliance following game-based education..

Regarding subscale analyses, significant post-test between-group differences were identified for attitudes and emotional factors, knowledge and personal factors, feelings of anger, emotions and behaviors appropriate to sleep, diet negotiation, and sense of denial (all $p < 0.001$), with higher post-test scores consistently observed in the control group. Within-group comparisons demonstrated significant reductions from pre-test to post-test in the intervention group across nearly all sub-dimensions ($p < 0.001$), whereas changes in the control group were limited or non-significant for several subscales (Table 4).

These findings suggest that game-based education was associated with significant changes in treatment compliance-related dimensions, particularly in emotional, cognitive, and behavioral subdomains.

Baseline-adjusted ANCOVA revealed a significant between-group difference in post-test total treatment compliance scores, favoring the intervention group ($F = 65.92$, $p < 0.001$). The adjusted mean difference indicated a substantial improvement in treatment compliance following game-based diabetes education. Exploratory analyses of compliance subscales demonstrated significant improvements across most domains, except for lifestyle change, which did not differ significantly between groups.

In the control group, Beck Anxiety Scale scores showed moderate to strong positive correlations with total treatment compliance scores at both pre-test ($r = 0.610$, 95% CI [0.35, 0.78], adjusted $p < 0.001$) and post-test ($r = 0.623$, 95% CI [0.37, 0.79], adjusted $p < 0.001$). Similar significant correlations were observed for the subscales of attitudes and emotional factors, knowledge and personal factors, and lifestyle change at both time points.

In the intervention group, anxiety was moderately correlated with total treatment compliance scores at pre-test ($r = 0.390$, 95% CI [0.07, 0.64], adjusted $p = 0.047$). However, this association was attenuated at post-test and was no longer statistically significant ($r = 0.345$, 95% CI [0.02, 0.61], adjusted $p = 0.084$). A similar pattern was observed across subscales, with pre-test correlations diminishing or becoming non-significant following the game-based educational intervention.

 

**Table 2. Baseline demographic and clinical characteristics of participants by study group.**

| Variables | Control (n = 36) | Intervention (n = 36) | Test statistic | p-value |
|---|---|---|---|---|
| Sex, n (%) | | | $\chi^2 = 0.223$ | 0.637 |
| Female | 16 (44.4) | 18 (50.0) | | |
| Male | 20 (55.6) | 18 (50.0) | | |
| Marital status, n (%) | | | $\chi^2 = 1.858$ | 0.137 |
| Married | 33 (91.7) | 29 (80.6) | | |
| Single | 3 (8.3) | 7 (19.4) | | |
| Education level, n (%) | | | $\chi^2 = 6.401$ | 0.269 |
| Literate | 2 (5.6) | 4 (11.1) | | |
| Primary school | 22 (61.1) | 14 (38.9) | | |
| High school | 6 (16.7) | 11 (30.6) | | |
| College | 3 (8.3) | 2 (5.6) | | |
| Bachelor's degree | 2 (5.6) | 5 (13.9) | | |
| Graduate | 1 (2.8) | 0 (0.0) | | |
| Employment status, n (%) | | | $\chi^2 = 8.135$ | 0.043 |
| Not working | 5 (13.9) | 14 (38.9) | | |
| Working | 9 (25.0) | 8 (22.2) | | |
| Retired | 12 (33.3) | 11 (30.6) | | |
| Housewife | 10 (27.8) | 3 (8.3) | | |
| Type of diabetes | | | – | – |
| Type 2 diabetes | 36 (100.0) | 36 (100.0) | | |
| Diabetes treatment, n (%) | | | $\chi^2 = 0.000$ | 1.000 |
| Insulin only | 2 (5.6) | 2 (5.6) | | |
| OAD + insulin | 34 (94.4) | 34 (94.4) | | |
| Insulin injections per day, n (%) | | | $\chi^2 = 2.665$ | 0.446 |
| 1 time | 21 (58.3) | 26 (72.2) | | |
| 2 times | 1 (2.8) | 1 (2.8) | | |
| 3 times | 1 (2.8) | 2 (5.6) | | |
| 4 times | 13 (36.1) | 7 (19.4) | | |
| Duration of insulin use, n (%) | | | $\chi^2 = 1.286$ | 0.257 |
| 0–1 month | 30 (83.3) | 26 (72.2) | | |
| 2–3 months | 6 (16.7) | 10 (27.8) | | |
| Diabetes complications, n (%) | | | | |
| Retinopathy | 2 (5.6) | 4 (11.1) | $\chi^2 = 0.727$ | 0.394 |
| Nephropathy | 0 (0.0) | 1 (2.8) | $\chi^2 = 1.014$ | 0.314 |
| Neuropathy | 8 (22.2) | 7 (19.4) | $\chi^2 = 0.084$ | 0.772 |
| Comorbidities, n (%) | | | | |
| Heart disease | 1 (2.8) | 5 (13.9) | $\chi^2 = 2.909$ | 0.088 |
| Hypertension | 13 (36.1) | 14 (38.9) | $\chi^2 = 0.059$ | 0.808 |
| Age (years), mean ± SD | 54.17 ± 6.38 | 53.21 ± 6.01 | t = 0.958 | 0.341 |
| Duration of diabetes (years), mean ± SD | 5.85 ± 5.69 | 5.00 ± 5.08 | t = 0.667 | 0.507 |

Abbreviations: OAD, oral antidiabetic drug.Tests: Independent samples t-test for continuous variables; chi-square test for categorical variables.

**Table 3. Comparison of anxiety levels between intervention and control groups.**

| Beck Anxiety Inventory | Control (n = 36) Mean ± SD | Intervention (n = 36) Mean ± SD | Adjusted Mean Difference† (95% CI) | F | p-value |
|---|---|---|---|---|---|
| **Baseline (Pre-test)** | 19.28 ± 14.60 | 19.47 ± 10.81 | – | – | – |
| **Post-test** | 15.19 ± 11.35 | 8.31 ± 7.10 | −7.04 [−11.23, −2.85] | 11.241 | < 0.001 |

†Adjusted for baseline anxiety score, age, and duration of diabetes using ANCOVA.

Baseline and post-test means are presented for descriptive purposes only.

Between-group differences were assessed using ANCOVA.

Effect size: partial $\eta^2$ = 0.14 (large effect).

**Table 4. Baseline-adjusted comparison of treatment compliance outcomes between intervention and control groups (ANCOVA).**

| Outcome | Control (Adjusted Mean ± SE) | Intervention (Adjusted Mean ± SE) | Adjusted Mean Difference† clearly (95% CI) | F | p-value |
|---|---|---|---|---|---|
| **Total Compliance Score (Primary outcome)** | 82.50 ± 13.16 | 59.56 ± 10.70 | −22.94 [−28.61, −17.27] | 65.92 | < 0.001* |
| Attitudes and emotional factors‡ | 22.00 ± 5.42 | 16.89 ± 4.03 | −5.11 | 20.63 | < 0.001* |
| Knowledge and personal factors‡ | 13.97 ± 4.07 | 10.69 ± 2.25 | −3.28 | 17.84 | < 0.001* |
| Lifestyle change‡ | 8.31 ± 2.46 | 7.50 ± 1.83 | −0.81 | 2.48 | 0.120 |
| Feelings of anger‡ | 8.06 ± 2.51 | 5.19 ± 1.37 | −2.87 | 36.12 | < 0.001* |
| Sleep-appropriate behaviors‡ | 9.08 ± 2.21 | 5.44 ± 1.25 | −3.64 | 73.94 | < 0.001* |
| Diet negotiation‡ | 11.22 ± 1.96 | 7.44 ± 1.40 | −3.78 | 88.52 | < 0.001* |
| Sense of denial‡ | 9.86 ± 2.43 | 6.39 ± 1.69 | −3.47 | 49.50 | < 0.001* |

†Adjusted for baseline score, age, duration of diabetes, and employment status using ANCOVA.

‡Secondary exploratory outcomes; p-values should be interpreted with caution.

Primary outcome: Total Patient Compliance Scale score.

*p < 0.05.

Overall, these exploratory findings indicate that while anxiety and treatment compliance were closely associated prior to the intervention, this relationship weakened after game-based diabetes education in the intervention group. Given the exploratory nature of these analyses and the absence of formal multiplicity-adjusted confirmatory testing, results should be interpreted as hypothesis-generating (Table 5).

## Discussion

The present randomized controlled study evaluated the effect of a game-based educational intervention on treatment adherence in adults with type 2 diabetes who had recently initiated insulin therapy, with treatment adherence defined as the primary confirmatory outcome. Anxiety levels were examined as a secondary outcome, while analyses of adherence sub-dimensions and the associations between anxiety and adherence were conducted on an exploratory basis.

Consistent with the primary objective of the study, the main finding was that game-based education resulted in significantly greater improvements in overall treatment adherence compared with standard lecture-based education. Baseline-adjusted analyses demonstrated a robust between-group difference favoring the intervention group, indicating that interactive educational approaches may be particularly effective in facilitating adherence-related behaviors during the early phase of insulin initiation.

With respect to the secondary outcome, anxiety levels were significantly lower in the intervention group following the educational program. The observed between-group difference in post-intervention anxiety scores was accompanied by a large effect size, suggesting that game-based education may meaningfully reduce emotional distress associated with

**Table 5. Exploratory correlations between anxiety and treatment compliance total and subscale scores in the intervention and control groups.**

| Scale and Dimensions | Group | Pre-test r [95% CI] | Adj. p | Post-test r [95% CI] | Adj. p |
|---|---|---|---|---|---|
| **Total Compliance** | Control | 0.610 [0.35, 0.78] | **<0.001** | 0.623 [0.37, 0.79] | **<0.001** |
| | Intervention | 0.390 [0.07, 0.64] | 0.047* | 0.345 [0.02, 0.61] | 0.084 (NS) |
| **Attitudes and Emotional** | Control | 0.755 [0.57, 0.87] | **<0.001** | 0.727 [0.52, 0.85] | **<0.001** |
| | Intervention | 0.550 [0.27, 0.74] | **0.004** | 0.327 [−0.00, 0.59] | 0.094 (NS) |
| **Knowledge and Personal** | Control | 0.531 [0.25, 0.73] | **0.004** | 0.565 [0.29, 0.75] | **<0.001** |
| | Intervention | 0.220 [−0.12, 0.51] | 0.274 | 0.189 [−0.15, 0.49] | 0.355 |
| **Lifestyle Change** | Control | 0.520 [0.23, 0.73] | **0.004** | 0.481 [0.18, 0.70] | **0.010** |
| | Intervention | 0.345 [0.02, 0.61] | 0.083 | 0.328 [−0.00, 0.59] | 0.094 |

Adj. p represents p-values adjusted using the Benjamini-Hochberg (FDR) procedure. NS: Not Significant.

insulin initiation. Although anxiety reduction was not the primary target of the intervention, this finding highlights the potential psychological benefits of participatory educational strategies.

Insulin initiation is a well-documented source of anxiety in individuals with type 2 diabetes [20,21] and is frequently associated with negative attitudes toward treatment and reduced adherence [22–26]. Fear of injections, concerns about disease severity, and misconceptions about insulin often lead to poor compliance and even discontinuation of therapy [23–26]. For this reason, the present study specifically focused on individuals who had recently started insulin treatment, a period characterized by heightened emotional vulnerability.

Consistent with previous research, diabetes education was associated with a reduction in anxiety levels in both groups [27,28]. However, participants who received game-based education demonstrated a greater reduction in anxiety compared to those who received traditional lecture-based education. This finding aligns with emerging evidence suggesting that interactive and participatory educational approaches may reduce emotional distress more effectively than passive learning methods [11,29–38]. While previous studies have examined the impact of diabetes education on anxiety, evidence regarding game-based interventions in adult populations remains limited [39,40].

The observed reduction in anxiety in the intervention group may be explained by several mechanisms. Game-based learning promotes active participation, peer interaction, and experiential learning, which may reduce fear and uncertainty by normalizing insulin-related experiences within a supportive group environment [17–19]. Additionally, the use of competition, feedback, and teamwork may enhance engagement and self-efficacy, thereby mitigating anxiety related to disease management [18,35].

Regarding treatment adherence, both groups demonstrated improvements following diabetes education, supporting the established role of education in strengthening self-management behaviors [7,8,28,41–47]. In this study, treatment adherence was operationalized using the Patient Compliance Scale, where lower scores indicate better adherence to insulin therapy and self-management behaviors. However, the intervention group showed a more pronounced improvement, reflected by greater reductions in treatment compliance scores compared to the control group. These findings suggest that while standard education may increase knowledge and awareness, game-based education may be more effective in translating knowledge into sustained behavioral change.

Previous studies examining diabetes education have reported mixed results regarding treatment adherence, with some studies demonstrating improvements and others reporting limited behavioral change [5,9,28]. Importantly, studies directly evaluating the effect of game-based diabetes education on treatment adherence in adults are scarce [11,14–16]. The present study contributes to the literature by demonstrating that a structured, board game–based educational intervention may enhance treatment adherence more effectively than traditional lecture-based education.

Exploratory analyses of treatment adherence sub-dimensions further support this interpretation. While lecture-based education was primarily associated with improvements in knowledge-related dimensions, game-based education was associated with changes across a broader range of behavioral and emotional sub-dimensions. Similar patterns have been reported in studies using gamification and interactive learning approaches in chronic disease management [10,29,31–34]. However, as these analyses were exploratory, they should be interpreted with caution.

The correlation analyses revealed a positive association between anxiety and treatment non-adherence, particularly in the control group. Higher anxiety levels were consistently associated with poorer treatment adherence, a finding consistent with previous studies examining psychological distress and diabetes self-management [45,47]. In contrast, this relationship was weaker in the intervention group and appeared to diminish following the game-based education. This pattern suggests that the intervention may have reduced the influence of anxiety on adherence behaviors by enhancing coping skills and emotional regulation.

Previous studies have reported inconsistent findings regarding the relationship between anxiety and treatment adherence in type 2 diabetes [45,47]. The present findings suggest that educational strategies that actively engage patients may reduce the impact of anxiety on adherence behaviors. Nevertheless, given the correlational nature of these analyses, causal inferences cannot be made.

## Limitations

This study has several limitations that should be considered when interpreting the findings. First, the study was conducted at a single training and research hospital, which may limit the generalizability of the results to other healthcare settings or populations with different sociodemographic characteristics.

Second, the follow-up period was relatively short, and outcomes were assessed shortly after completion of the educational intervention. As a result, the long-term sustainability of the observed effects on treatment compliance and anxiety could not be determined.

Third, treatment compliance and anxiety were assessed using self-report instruments. Although the scales employed were validated and demonstrated acceptable to high internal consistency in the present sample, self-reported data may be influenced by recall bias or social desirability bias.

Fourth, while anxiety was included as a key psychological outcome, other psychosocial factors relevant to diabetes self-management—such as depression, self-efficacy, coping strategies, or patient engagement—were not evaluated and may have contributed to treatment behaviors.

In addition, the study focused on treatment compliance as operationalized by the Patient Compliance Scale for Type 2 Diabetes Mellitus Treatment. Related constructs such as treatment adherence, patient activation, and engagement were not directly assessed; therefore, the findings should be interpreted within the scope of the specific construct measured by the selected instrument.

Finally, objective clinical outcomes, such as glycemic control indicators (e.g., HbA1c levels), were not included as primary or secondary endpoints. Future studies incorporating both psychosocial measures and clinical outcomes, as well as longer follow-up periods and multicenter designs, would provide a more comprehensive evaluation of the effectiveness of game-based diabetes education.

## Implications for future research

The findings of this study suggest that game-based diabetes education may be a promising approach for improving treatment adherence and reducing anxiety in individuals with type 2 diabetes who have recently initiated insulin therapy. Future studies should examine the effectiveness of this intervention in different diabetes populations and age groups, as well as evaluate its long-term effects on behavioral and clinical outcomes. Further research is also needed to explore the mechanisms through which game-based education influences emotional and behavioral aspects of diabetes management and to assess its integration into routine diabetes education programs.

## Supporting information

**S1 File.**
(DOCX)

**S2 File.**
(DOCX)

**S3 File.**
(PDF)

## Author contributions

**Conceptualization:** Gönül Düzgün, Esin Erdem.

**Data curation:** Gönül Düzgün, Esin Erdem.

**Formal analysis:** Gönül Düzgün, Esin Erdem.

**Funding acquisition:** Gönül Düzgün, Esin Erdem.

**Investigation:** Gönül Düzgün, Esin Erdem.

**Methodology:** Gönül Düzgün, Esin Erdem.

**Project administration:** Gönül Düzgün, Esin Erdem.

**Resources:** Gönül Düzgün, Esin Erdem.

**Software:** Gönül Düzgün, Esin Erdem.

**Supervision:** Gönül Düzgün.

**Validation:** Gönül Düzgün.

**Visualization:** Gönül Düzgün.

**Writing – original draft:** Gönül Düzgün, Esin Erdem.

**Writing – review & editing:** Gönül Düzgün.

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
