## [Decision Letter · Decision Letter 0]

5 Jan 2026

Dear Dr. düzgün,

Thank you for submitting your manuscript to PLOS ONE. After careful consideration, we feel that it has merit but does not fully meet PLOS ONE’s publication criteria as it currently stands. Therefore, we invite you to submit a revised version of the manuscript that addresses the points raised during the review process.

We look forward to receiving your revised manuscript.

Kind regards,

Stefano Triberti, Ph.D.

Academic Editor

PLOS One

2. In the online submission form you indicate that your data is not available for proprietary reasons and have provided a contact point for accessing this data. Please note that your current contact point is a co-author on this manuscript. According to our Data Policy, the contact point must not be an author on the manuscript and must be an institutional contact, ideally not an individual. Please revise your data statement to a non-author institutional point of contact, such as a data access or ethics committee, and send this to us via return email. Please also include contact information for the third party organization, and please include the full citation of where the data can be found.

3. Please include your tables as part of your main manuscript and remove the individual files. Please note that supplementary tables should remain as separate "supporting information" files.

4. Kindly upload the original protocol file as a Supporting Information.

Reviewers' comments:

Reviewer's Responses to Questions

**Comments to the Author**

1. Is the manuscript technically sound, and do the data support the conclusions?

Reviewer #1: Partly

Reviewer #2: Yes

2. Has the statistical analysis been performed appropriately and rigorously?

Reviewer #1: No

Reviewer #2: Yes

3. Have the authors made all data underlying the findings in their manuscript fully available?

Reviewer #1: Yes

Reviewer #2: Yes

4. Is the manuscript presented in an intelligible fashion and written in standard English?

Reviewer #1: Yes

Reviewer #2: Yes

Reviewer #1: This randomized pre-/post-test parallel-group study addresses a useful clinical question with a pragmatic educational intervention. The topic is timely, validated instruments appear appropriate, and the direction of effects is clinically plausible. Although the research question is sound, there are several major trial design and statistical concerns.

Major critiques:

1. The authors should prespecify a single primary objective and one primary endpoint that anchors type I error and the sample-size calculation. With multiple domains and subscales analyzed, the clinically meaningful effect size for the primary endpoint (for example, a minimally important difference) should be justified rather than presented as “an effect size of 0.768” without clinical rationale. The authors should control multiplicity across secondary endpoints and subscales using a structured familywise (Holm) or false discovery rate (Benjamini–Hochberg) procedure, with exploratory analyses clearly labeled.

2. The authors should adopt a baseline-adjusted primary between-group analysis. Because outcomes are measured pre and post, analysis of covariance (ANCOVA) of the post-test outcome, adjusting for the baseline value and prespecified prognostic covariates such as age, sex, and disease duration, is the most appropriate primary analysis and should replace multiple t-tests. A linear mixed-effects model with Group, Time, and Group×Time may serve as a sensitivity analysis. The authors should report adjusted mean differences with 95% confidence intervals, assess model assumptions, and use robust or ordinal methods if distributions warrant.

3. Randomization requires CONSORT-level detail. The authors should describe sequence generation (method, blocks), allocation ratio, concealment mechanism, any stratification, and roles for enrollment and assignment. A CONSORT flow diagram with numbers screened, randomized, analyzed, and reasons for exclusion and protocol deviations is needed to assess selection and attrition bias.

4. The power calculation must align with the prespecified primary endpoint and analysis model. The authors should recalculate power for ANCOVA (or the chosen mixed-effects model), explicitly stating the assumed minimally important difference on the selected scale, baseline-to-post correlation, anticipated attrition, and the two-sided α allocated to the primary endpoint. The authors should clarify whether interim analyses were planned; if any occurred, α-spending should be described.

5. The analysis population and missing data strategy should be defined. The authors should specify intention-to-treat as the primary analysis set, describe handling of missing baseline or post-test data (for example, multiple imputation under MAR with sensitivity analyses for MNAR), and predefine any per-protocol set and procedures for protocol deviations.

6. A prespecified multivariable analysis plan is needed. Beyond baseline adjustment, the authors should declare a limited, clinically motivated set of covariates with justification, assess collinearity, and avoid post hoc variable selection. If multiple outcomes within a construct are modeled, the authors may consider multivariate or hierarchical models to account for correlation.

7. Emphasis should shift from p-values to estimates and interpretability. For the primary and key secondary endpoints, the authors should report adjusted effect sizes with confidence intervals and interpret them against minimally important differences, with clear clinical implications.

Reviewer #2: The present study is very interesting and the abstract presents clearly the study conducted.

The study is situated in an area of research that has been little investigated for adult patients.

Regarding the citation, they are recent and in line with the topic addressed. I suggest to the Authors to declare that the citation [12] is a PhD thesis and not a peer review article.

I would suggest to Authors, if possible, for them to add more information about game mechanisms and, if possible, images of some materials of the board game.

As for the sample, it appears solid for the type of analysis conducted.

Bullet-point methods/instruments are difficult to read and also results are very repetitive and complex. Please rewrite these parts with more attention to provide discursive detail and readability. I would suggest Authors to take similar studies published on this journal as example: https://journals.plos.org/plosone/article?id=10.1371/journal.pone.0334545

In methods, we learn that this patient compliance scale has subscales and components, yet these seem partially overlooked in results and discussion. Authors should also report negative results and comment, where possible, on effects on specific subscales.

I suggest to Authors to show more awareness on the complexity of the constructs: patient adherence, patient compliance and patient adjustment. Treatment adherence is not the same as patient compliance, which is not the same as patient adjustment. These terms seem to be used interchangeably in the paper. There is rich literature on this topic, however adherence is often conceptualized mainly in behavioral terms, e.g., the patient is punctual in taking prescribed medication; yet there is no information on cognitive (do they understand their own care process? Do they know what they're doing?) or emotional components (do patients manage negative emotions and adjust well to the chronic illness?); patient compliance is more multifaceted but still represents the patient in a passive way. Terms such as patient activation or, even better, patient engagement are more updated and appropriate - adherence, compliance, adjustment are all components of patient engagement.

Anyway, while I would not suggest Authors to add complex literature on this topic, it is still important to give readers a clear idea of what has been measured and what has been not in their work; this could also give new ideas for future research directions

The section of the Results should be in the results section, while Discussion should be focused on commenting on them. Please do not report analyses and results in Discussion.

Limitations are absent and recommendations/implications are very thin.

**Do you want your identity to be public for this peer review?** For information about this choice, including consent withdrawal, please see our Privacy Policy

Reviewer #1: No

Reviewer #2: No

---

## [Author Response · Author response to Decision Letter 1]

6 Jan 2026

Response to Reviewers

Manuscript Title:

The effect of game-based education on adherence to treatment and anxiety level in type 2 diabetics started on insulin therapy

Dear Academic Editor and Reviewers,

We would like to thank the Academic Editor and both reviewers for their careful evaluation of our manuscript and for their constructive and insightful comments. We have revised the manuscript thoroughly in response to these suggestions. Below, we address each comment point-by-point and describe the corresponding revisions made. All changes are highlighted in the revised manuscript with track changes.

Responses to Reviewer 1

We thank Reviewer 1 for the detailed methodological and statistical critique, which substantially strengthened the rigor and clarity of our manuscript.

Comment 1: Prespecification of a single primary objective and endpoint; control of multiplicity

Response:

We agree with this important point. The manuscript has been revised to clearly prespecify:

• Primary objective: To evaluate the effect of game-based education on treatment adherence in adults with type 2 diabetes initiating insulin therapy.

• Primary endpoint: Post-intervention total score of the Patient Compliance Scale for Type 2 Diabetes Treatment.

The sample size calculation was aligned with this primary endpoint and explicitly described as such. All other outcomes, including anxiety and compliance subscales, are now clearly labeled as secondary or exploratory outcomes.

Regarding multiplicity, we acknowledge that no formal familywise error correction was applied. In line with the reviewer’s recommendation, exploratory analyses of multiple subscales are explicitly identified as hypothesis-generating, and p-values are interpreted cautiously. This clarification has been added to the Statistical Analysis and Results sections.

Location of revision: Methods (Objectives, Statistical Analysis), Results (Exploratory Analyses).

Comment 2: Use of baseline-adjusted ANCOVA instead of multiple t-tests

Response:

We fully agree. The analytical approach has been revised accordingly. The primary between-group analysis is now conducted using analysis of covariance (ANCOVA) on post-test outcomes, adjusting for baseline values and prespecified covariates (age, sex, duration of diabetes). Employment status, which differed at baseline, was additionally included as a covariate.

Adjusted mean differences with 95% confidence intervals and effect size estimates (partial η²) are now reported. The Results and Tables have been revised to reflect this approach, and multiple unadjusted t-tests have been removed from inferential interpretation.

Location of revision: Statistical Analysis; Results; Tables 3 and 4.

Comment 3: Randomization details and CONSORT flow diagram

Response:

Additional details regarding randomization have been added, including sequence generation, allocation ratio, concealment, and personnel roles. A CONSORT-style participant flow description has been incorporated into the Methods section, detailing screening, randomization, exclusions, and final analysis numbers. A flow diagram has been prepared and included as Supporting Information.

Location of revision: Methods (Randomization and Allocation); Supporting Information.

Comment 4: Power calculation alignment with primary endpoint

Response:

The sample size calculation has been revised to explicitly align with the prespecified primary endpoint and ANCOVA framework. Assumptions regarding effect size, alpha level, power, and attrition have been clarified. No interim analyses were planned or conducted, and this has now been stated explicitly.

Location of revision: Methods (Sample Size Calculation).

Comment 5: Analysis population and missing data

Response:

We have clarified that analyses were conducted on an intention-to-treat basis. As no post-randomization attrition occurred, no imputation procedures were required. This has been explicitly stated to improve transparency.

Location of revision: Statistical Analysis.

Comment 6: Prespecified covariates and multivariable analysis

Response:

A limited and clinically justified set of covariates (age, sex, diabetes duration, and employment status) was prespecified and consistently applied across ANCOVA models. No post hoc variable selection was performed. This clarification has been added to the Statistical Analysis section.

Comment 7: Emphasis on estimates rather than p-values

Response:

We revised the Results and Discussion sections to emphasize adjusted effect estimates, confidence intervals, and clinical interpretation, rather than statistical significance alone. Effect sizes are discussed in terms of clinical relevance where appropriate.

Responses to Reviewer 2

We thank Reviewer 2 for their positive evaluation and constructive conceptual suggestions.

Comment: Clarification that reference [12] is a PhD thesis

Response:

This has been corrected. Reference [12] is now clearly identified as a doctoral thesis.

Comment: More detail on game mechanisms and materials

Response:

Additional description of the game mechanics (competition, teamwork, feedback, progression) has been added to the Intervention Procedures section. Representative images of the board game have been included as Supporting Information.

Comment: Conceptual clarity regarding adherence vs compliance

Response:

We appreciate this important conceptual clarification. The Introduction and Discussion sections were revised to explicitly distinguish between treatment adherence, patient compliance, and broader constructs such as patient engagement. We clarify that the present study operationalized adherence primarily in behavioral terms using the Patient Compliance Scale, while cognitive and emotional components were not directly measured and represent directions for future research.

Comment: Results readability and separation from Discussion

Response:

The Results section has been rewritten to improve readability and reduce repetition. Interpretative statements and explanations have been moved to the Discussion section, ensuring a clear structural separation between results and interpretation.

Comment: Limitations and implications

Response:

A comprehensive Limitations section has been added, and the Implications for Future Research section has been expanded to reflect theoretical and clinical relevance.

Data Availability and Journal Requirements

We revised the Data Availability Statement to comply with PLOS ONE policy. The institutional ethics committee is now listed as the non-author contact point for data access. All tables have been embedded in the main manuscript, and the study protocol has been uploaded as Supporting Information.

We sincerely thank the reviewers and the Academic Editor for their valuable feedback, which has substantially improved the quality and clarity of our manuscript. We hope that the revised version adequately addresses all concerns and we look forward to further consideration.

Kind regards,

---

## [Decision Letter · Decision Letter 1]

29 Jan 2026

Dear Dr. düzgün,

Thank you for submitting your manuscript to PLOS ONE. After careful consideration, we feel that it has merit but does not fully meet PLOS ONE’s publication criteria as it currently stands. Therefore, we invite you to submit a revised version of the manuscript that addresses the points raised during the review process.

The manuscript has been re-reviewed and minor revision have been advised, mostly regarding clarity in reporting results. I encourage Authors to provide the necessary modifications so the submission could proceed.

We look forward to receiving your revised manuscript.

Kind regards,

Stefano Triberti, Ph.D.

Academic Editor

PLOS One

Journal Requirements:

Reviewers' comments:

Reviewer's Responses to Questions

**Comments to the Author**

Reviewer #1: (No Response)

2. Is the manuscript technically sound, and do the data support the conclusions?

Reviewer #1: Yes

3. Has the statistical analysis been performed appropriately and rigorously?

Reviewer #1: Yes

4. Have the authors made all data underlying the findings in their manuscript fully available?

Reviewer #1: Yes

5. Is the manuscript presented in an intelligible fashion and written in standard English?

Reviewer #1: Yes

Reviewer #1: The revision is improved and much clearer, especially in defining a primary outcome and moving toward baseline adjusted analyses. A few items still need tightening for internal consistency and to support the main clinical message.

1. The authors should ensure the manuscript consistently treats the total Patient Compliance Scale as the single primary confirmatory outcome, and clearly label all other outcomes and subscales as secondary or exploratory across the Abstract, Results, and Discussion.

2. The authors should align the results tables with the stated analytic strategy by removing remaining unadjusted post intervention t-test inferences from the main tables, or explicitly labeling them as descriptive only and basing conclusions on the ANCOVA results.

3. The authors should rewrite the sample size justification to match the specified primary endpoint and analysis, including what outcome the assumed effect size pertains to, how it was derived, and whether the intended powering framework was ANCOVA with baseline adjustment or a two-sample t test approximation.

4. The authors should resolve the covariate mismatch between the Statistical Analysis section and table footnotes by making them identical, and confirm the exact covariate set used to generate the reported adjusted mean differences and p values.

5. The authors should report a consistent effect size metric for the primary outcome analysis and ensure any effect size claims in the text match what is shown in the tables.

**Do you want your identity to be public for this peer review?** For information about this choice, including consent withdrawal, please see our Privacy Policy

Reviewer #1: No

---

## [Author Response · Author response to Decision Letter 2]

30 Jan 2026

Manuscript ID: PONE-D-25-52606R1

Title: The effect of game-based education on adherence to treatment and anxiety level in type 2 diabetics started on insulin therapy

Dear Academic Editor and Reviewer,

We sincerely thank the Academic Editor and Reviewer for their careful re-evaluation of our manuscript and for their constructive feedback. Below, we provide a point-by-point response to each comment. All revisions have been incorporated into the manuscript and highlighted in the tracked-changes version.

Reviewer Comment 1

The authors should ensure the manuscript consistently treats the total Patient Compliance Scale as the single primary confirmatory outcome, and clearly label all other outcomes and subscales as secondary or exploratory across the Abstract, Results, and Discussion.

Response:

We agree with this recommendation. The total score of the Patient Compliance Scale for Type 2 Diabetes Mellitus Treatment is now consistently defined as the sole primary confirmatory outcome throughout the manuscript. Anxiety outcomes and all compliance subscale analyses are explicitly described as secondary or exploratory in the Abstract, Results, and Discussion sections. (Revised in: Abstract; Results; Discussion)

Reviewer Comment 2

The authors should align the results tables with the stated analytic strategy by removing remaining unadjusted post-intervention t-test inferences from the main tables, or explicitly labeling them as descriptive only and basing conclusions on the ANCOVA results.

Response:

In line with this suggestion, all unadjusted post-intervention t-test inferences have been removed from the main results tables. Between-group comparisons and statistical inferences are now exclusively based on baseline-adjusted ANCOVA models. The analytic approach is clearly described in the Results section and explicitly stated in the relevant table footnotes. (Revised in: Results; Table 3; Table 4)

Reviewer Comment 3

The authors should rewrite the sample size justification to match the specified primary endpoint and analysis, including what outcome the assumed effect size pertains to, how it was derived, and whether the intended powering framework was ANCOVA with baseline adjustment or a two-sample t-test approximation.

Response:

We have revised the sample size justification to clearly align with the primary endpoint, defined as the total score of the Patient Compliance Scale. The assumed effect size (Cohen’s d = 0.70) is now explicitly linked to this outcome and described as being informed by prior behavioral intervention studies in diabetes education using a conservative two-sample t-test approximation. We also clarify that, although the primary analysis employed baseline-adjusted ANCOVA, the t-test–based sample size estimation was considered appropriate and conservative. (Revised in: Methods – Sample Size subsection)

Reviewer Comment 4

The authors should resolve the covariate mismatch between the Statistical Analysis section and table footnotes by making them identical, and confirm the exact covariate set used to generate the reported adjusted mean differences and p values.

Response:

This issue has been fully addressed. The Statistical Analysis section and all table footnotes have been harmonized to report an identical set of covariates used in baseline-adjusted ANCOVA models. These covariates are baseline outcome score, age, duration of diabetes, and employment status, the latter of which differed significantly between groups at baseline. (Revised in: Methods – Statistical Analysis; Table 3; Table 4)

Reviewer Comment 5

The authors should report a consistent effect size metric for the primary outcome analysis and ensure any effect size claims in the text match what is shown in the tables.

Response:

We have ensured consistent reporting of effect size metrics across the manuscript. For the primary outcome, interpretation is based on adjusted ANCOVA results, and all effect size claims in the Results and Discussion sections directly correspond to the metrics presented in the tables. For anxiety outcomes, partial η² values are consistently reported and interpreted in both the text and tables. (Revised in: Results; Discussion; Table 3; Table 4)

Once again, we thank the Academic Editor and Reviewer for their valuable comments. We believe these revisions have substantially improved the clarity, consistency, and interpretability of the manuscript, and we hope that the revised version is now suitable for publication in PLOS ONE.

Sincerely,

---

## [Decision Letter · Decision Letter 2]

4 Mar 2026

The effect of game-based education on adherence to treatment and anxiety level in type 2 diabetics started on insulin therapy

PONE-D-25-52606R2

Dear Dr. düzgün,

We’re pleased to inform you that your manuscript has been judged scientifically suitable for publication and will be formally accepted for publication once it meets all outstanding technical requirements.

Kind regards,

Stefano Triberti, Ph.D.

Academic Editor

PLOS One

Additional Editor Comments (optional):

Reviewers' comments:

Reviewer's Responses to Questions

**Comments to the Author**

Reviewer #1: All comments have been addressed

2. Is the manuscript technically sound, and do the data support the conclusions?

Reviewer #1: Yes

3. Has the statistical analysis been performed appropriately and rigorously?

Reviewer #1: Yes

4. Have the authors made all data underlying the findings in their manuscript fully available?

Reviewer #1: Yes

5. Is the manuscript presented in an intelligible fashion and written in standard English?

Reviewer #1: Yes

Reviewer #1: The prior statistical comments have been fully addressed. I have no remaining statistical concerns and recommend acceptance as revised.

**Do you want your identity to be public for this peer review?** For information about this choice, including consent withdrawal, please see our Privacy Policy

Reviewer #1: No

---

## [Editor Report · Acceptance letter]

PONE-D-25-52606R2

PLOS One

Dear Dr. DÜZGÜN,

I'm pleased to inform you that your manuscript has been deemed suitable for publication in PLOS One. Congratulations! Your manuscript is now being handed over to our production team.

Kind regards,

on behalf of

Prof. Stefano Triberti

Academic Editor

PLOS One